# Design and Analysis of the Self-Biased PLL with Adaptive Calibration for Minimum of the Charge Pump Current Mismatch

Xueming Wei [1,*], Renchuan Yin [1], Lingli Hou [2], Weilin Xu [1] and Baolin Wei [1]

[1] Guangxi Key Laboratory of Wireless Wideband Communication and Signal Processing, Guilin University of Electronic Technology, Guilin 541004, China; yrc521ic@163.com (R.Y.); xwl@guet.edu.cn (W.X.); guilinwxb@163.com (B.W.)
[2] Chengdu Sino Microelectronics Technology Co., Ltd., Chengdu 610041, China; ll_hou@csmsc.com
* Correspondence: scuweixue@guet.edu.cn

**Abstract:** A digital adaptive mismatch calibration (DAMC) circuit is proposed to decrease the output jitter of phase-locked loop (PLL). After amplifying the phase error with a linear time amplifier (TA), the DAMC adopts a successive approximation pulse width calibration method to reduce the mismatch current of the charge pump. The PLL prototype is fabricated in a 40nm process, the static phase error of the proposed PLL can be reduced from 358 ps to 10 ps at a 50 MHz reference clock approximately, and the RMS jitter of the PLL output is reduced from 4.91 ps to 3.59 ps, and the extended DAMC area only occupies 1.3% of the whole PLL area.

**Keywords:** current compensation; time amplifier; digital mismatch calibration; PLL

## 1. Introduction

Owing to the rapid development of semiconductor technology, the demand for low-jitter, high-frequency, and high-stability PLLs, which are used in various electronic systems, is increasing [1]. To decrease the output jitter, the bandwidth of the PLL must be changed significantly with respect to the input reference clock frequency. In particular, J.G. Maneatis designed a self-biased PLL [2], which has been extensively investigated because its wide bandwidth is independent of the process, voltage, temperature, or other parameters [3].

However, owing to the mismatch current of the charge pump in the phase-locked loop, the output spectrum of the PLL produces a large spurious at integer multiples of the reference clock frequency, the output noise performance of the PLL is significantly degraded, and some technologies have been presented to reduce the CP mismatch. Digital PLLs [4–6] are used to reduce charge leakage but require high-performance noise cancellation techniques. Analog mismatch current compensation circuits based on time digital conversion (TDC) have been proposed [7]; however, high-accuracy TDC also exhibits large power consumption. A differential CP with common-mode feedback has been proposed [8]. However, the structure in Ref. [8] required an additional operational amplifier. The operational amplifier has some nonideal effects, which may aggravate the performance of the PLL. The work in Ref. [9] used pulse injection in parallel to obtain the CP current mismatch error. However, the thermal noise and mismatch of capacitors lead to calibration errors for error accumulation owing to the use of many capacitors. In Ref. [10], the digital calibration method focused on improving the calibration accuracy; it kept the phase error time constant, but the phase-frequency detector (PFD) reset delay increased; however, the dead zone is still a difficult problem to solve.

In this study, a digital adaptive mismatch calibration circuit was proposed to compensate for the CP mismatch current. By increasing the phase error while keeping the PFD reset delay constant with a time amplifier (TA), the proposed design can easily detect and

compensate for dead zones. In contrast to the existing literature using TDC to calculate the pulse, the proposed scheme uses successive approximation (SAR) logic to compensate for the mismatch current and directly uses the phase error pulse signal as the clock control signal.

## 2. Proposed Architecture of PLL with DAMC

As shown in Figure 1, the proposed block diagram of the self-biased PLL with digital adaptive mismatch calibration (DAMC) consists of PFD, two identical charge-pump (CP1 and CP2) blocks, voltage-controlled oscillator (VCO), multi-mode N frequency divider (DIV), and DAMC circuit. All the voltage biases of the PLL are generated by the additional internal and no bandgap reference voltage source. The charge pump current is set to multiplied x of buffer bias current, which is given by [11]:

$$\frac{\omega_n}{\omega_{ref}} = \frac{1}{4\pi}\sqrt{\frac{xN}{\pi}}\sqrt{\frac{C_{vco}}{C}} \tag{1}$$

where N is the clock multiplication factor, $\omega_{ref}$ is the reference clock frequency, $\omega_n$ is the bandwidth of the PLL, $C$ is the filter capacitance, and $C_{VCO}$ is the equivalent VCO capacitance. Although the self-biased PLL is independent of process, voltage, temperature, or other parameters, as well as its ability to track the input clock frequency variation, it always results in mismatch current of the CP.

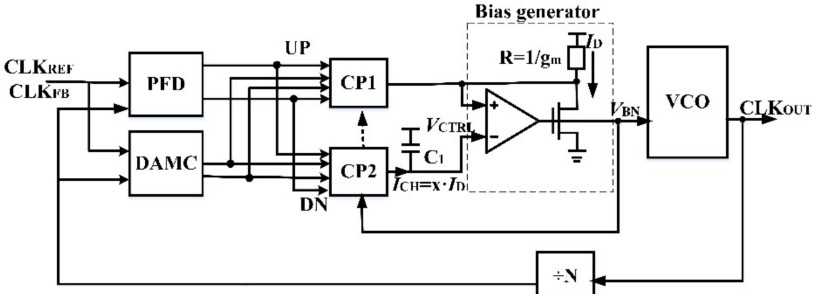

**Figure 1.** The proposed block diagram of the self-biased PLL with DAMC.

As shown in Figure 2, $V_{CTRL}$ is output voltage of the CP, and the differential architecture of the CP in the self-biased PLL results in a mismatched current at different $V_{CTRL}$. The current of the CP is determined by the feedback signal from the self-biased circuit. $V_{BN}$ is coming from the self-biased circuit. DN and UP are the output signals of the PFD, and DNB and UPB are their reverse signals. When the UP signal is high (UPB is low) and DN is low (DNB is high), the left branch of the differential structure is turned on, resulting in a charge pump charge. When the UP signal is low (UPB is high) and DN is high (DNB is low), the right branch of the differential structure is turned on, and the charge pump is discharged. When UP is high and DN is high, both the charge and discharge branches turn on, and, if there is no charge pump current mismatch, the output net charge is 0. However, the charge pump charge and discharge currents are usually not equal, resulting in a current mismatch.

It is assumed that the current $I_U$ flows from the supply to $V_{CTRL}$, and $I_D$ flows from $V_{CTRL}$ to the ground. Figure 2b shows the waveforms of the mismatch current at different $V_{CTRL}$. The variety of the $V_{CTRL}$ results in a mismatch between $I_D$ and $I_U$.

$$\Delta I_{CP} = I_U - I_P \tag{2}$$

The VCO waveform can be expressed as

$$V_{out} = V_0 \cos[w_0 t + K_{VCO}\int g(t)dt + K_{VCO}\int V_{CTR}dt] \tag{3}$$

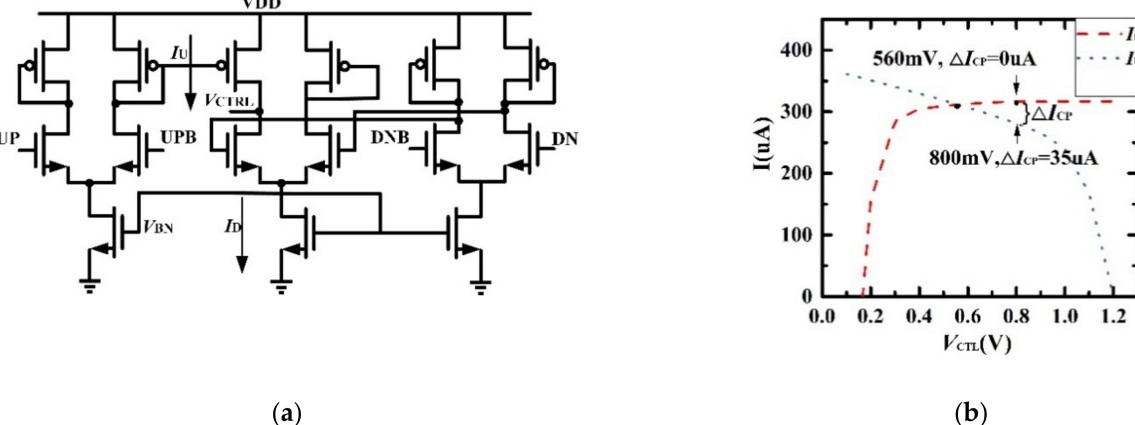

(**a**)                                                                                                                  (**b**)

**Figure 2.** Charge pump (**a**) circuit diagram; (**b**) mismatch current at different output voltage.

Where $V_0$ represents the oscillation amplitude of VCO, $V_{CTRL}$ represents the control voltage of VCO, $g(t)$ represents the change in $V_{CTRL}$ due to the charge pump mismatch. Figure 3 shows the changed value of $V_{CTRL}$ owing to the current mismatch. $g(t)$ is a triangle wave that performs a Fourier series expansion as follows:

$$g(t) = \frac{\Delta V \times \Delta t}{2T_{REF}} + \sum_{n=0}^{\infty} a_n \cos(nW_{REF} + \theta_n) \tag{4}$$

where $\Delta t$ represents the delay time owing to the charge pump current mismatch, $T_{REF}$ represents the reference signal cycle time, $W_{REF}$ represents the reference clock frequency. Further, in Figure 3, the corresponding waveforms of $V_{CTRL}$ are plotted, and the control voltage ripple is described as

$$\Delta V = \frac{\Delta I_{CP}}{C} T_d \tag{5}$$

where $T_d$ is the reset delay time of the PFD. By substituting (4) into (3), $V_{out}$ can be described as

$$V_{out} = V_0 [\cos(Wt) + \frac{K_{VCO} a_1}{2W_{REF}} \cos(W + W_{REF}) + \frac{K_{VCO} a_1}{2W_{REF}} \cos(W - W_{REF})] \tag{6}$$

where $a_1$ denotes Fourier series expansion coefficient of $g(t)$, and

$$W = W_0 + \frac{\Delta V \cdot \Delta t}{2T_{REF}} + K_{VCO} V_{CTRL} \tag{7}$$

The disturbance frequency component of $V_{out}$ is derived as

$$\Delta W = \frac{\Delta V \cdot \Delta t}{2T_{REF}} \tag{8}$$

The correspondence between mismatch current and phase error is as follows [10]:

$$\frac{\Delta I_{CP}}{I_{CP}} = \frac{\Delta t}{\Delta t + T_d} \tag{9}$$

Then,

$$\Delta V \cdot \Delta t = \frac{\Delta I_{CP}^2 \cdot T_d (\Delta t + T_d)}{C \cdot I_{CP}} \tag{10}$$

When $\Delta t \ll T_d$,

$$\Delta V \cdot \Delta t \approx \frac{\Delta I_{CP}^2 \cdot T_d^2}{C \cdot I_{CP}} \tag{11}$$

When $\Delta t \approx T_\mathrm{d}$, mismatch current is big,

$$\Delta V \cdot \Delta t \approx \frac{2\Delta I_{\mathrm{CP}}{}^2 \cdot T_\mathrm{d}{}^2}{C \cdot I_{\mathrm{CP}}} \tag{12}$$

It is known that $\Delta V \cdot \Delta t$ is proportional to $\Delta I_{\mathrm{cp}}{}^2$, and the perturbation frequency components $\Delta W$ of $V_{\mathrm{out}}$ are are proportional to $\Delta I_{\mathrm{cp}}{}^2$ from (11) and (12). The static phase error of the PFD is derived as

$$\Delta \varphi_{\mathrm{err}} = 2\pi \frac{\Delta t}{T_{\mathrm{REF}}} \times \frac{\Delta I_{\mathrm{CP}}}{I_{\mathrm{CP}}} \tag{13}$$

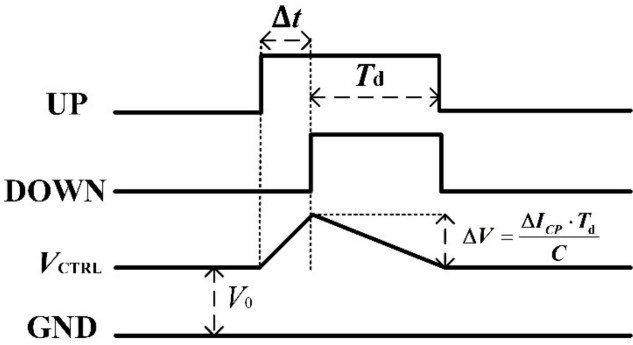

**Figure 3.** The changed value of $V_{\mathrm{CTRL}}$ due to current mismatch.

## 3. Circuit Design

The self-biased PLL with a DAMC was proposed to solve these problems. Figure 4 shows the current calibration circuit for generating the mismatch current compensation pulses: where one pulse (VDN) is generated because of the lagging reference clock, and another pulse (VUP) is generated because of the leading reference clock.

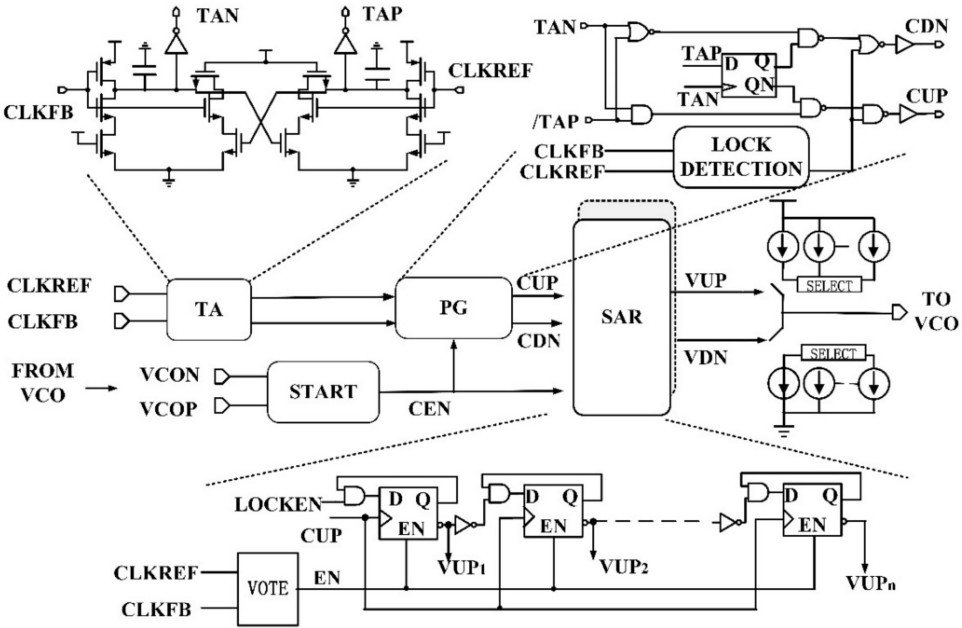

**Figure 4.** The digital adaptive mismatch calibration circuit.

Each PWC pulse is generated by a time amplifier (TA), pulse generator (PG), and two identical digitally successive approximation register (SAR) controlled logic. To further improve the calibration resolution, a time amplifier with reasonable accuracy, good linearity, and good linearity, and low power consumption was used. The time amplifier (TA) amplifies

the static reference phase error (after lock) to two times. The pulse generator (PG) circuit generates amplified reference phase error pulses (CUP and CDN).

The pulse generator (PG) circuit generates amplified reference phase-error pulses (CUP and CDN). A charge current source array and a discharge current source array were used to compensate for the mismatch current. Two pulse control signals $\text{VUP}_i(i = 1 \ldots n)$ and $\text{VDN}_i(i = 1 \ldots n)$ generated by the SAR control the charge and discharge current sources, respectively. After the SAR output pulse turned on the current array switch, the static phase error was measured again. The measured static phase error is processed by the DAMC, and then the current array switch is processed by the SAR output pulse signal again. Compensation is stopped when the static phase error cannot be measured by DAMC.

Figure 4 shows a simplified TA schematic used to amplify the reference phase error after the PLL locks. It consists of two cross coupling delay stages, where each stage controls the delay time of the other.

When the rising edge of the feedback clock (CLKFB) is behind the rising edge of the reference clock (CLKREF), CUP is generated by PG. The operation timing diagram of DAMC is shown in Figure 5. If the phase error is not amplified, the extracted pulse width is $\Delta t$. After the reference phase error is amplified, the extracted pulse width is $G \cdot \Delta t$, where $G$ is the amplification gain of TA.

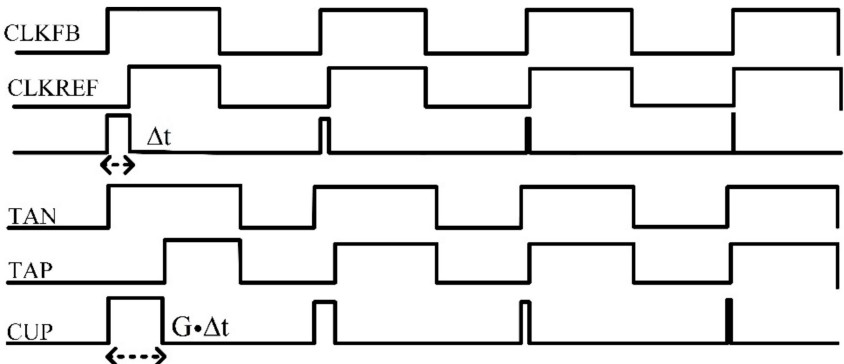

**Figure 5.** Phase error increase and extraction.

The static phase error decreased during the calibration process. When it is small enough (into dead zones), the pulse width will not be extracted, and the calibration will stop. TA amplified the pulse width, which contributed to the extraction of a smaller pulse width. The calibration accuracy of the DAMC ($A_{\text{cal}}$) is

$$A_{\text{cal}} = \frac{\Delta t_{\min}}{\Delta t_{\min} + G \cdot T_{\text{d}}} \tag{14}$$

The minimum detection time error of the DAMC, $\Delta t_{\min}$, is equivalent to the establishment time of the flip-flop. According to (13), the minimum inference phase error of the PFD, $\Delta \varphi_{\text{errmin}}$, can be described as

$$\Delta \varphi_{\text{errmin}} = 2\pi \frac{\Delta t^2_{\min}}{T_{\text{REF}}(\Delta t_{\min} + G \cdot T_d)} \tag{15}$$

From (14) and (15), using TA improves calibration accuracy.

Figure 6 shows the flowchart for DAMC. When the phase error was less than the minimum detection error of the DAMC, the calibration was stopped. This implies that the mismatch currents $\Delta I_{\text{CPi}}(i = 1 \ldots M)$ are minimized in M cycles. Figure 7 depicts the current-mismatch compensation example of a 5 bit compensation current source array. The minimum compensation current $\Delta I_{\min}$ was set to 1 µA, so, when compensation was completed, the CP mismatch current was less than 1 µA. The maximum mismatch current that can be compensated by the 5bit current source array is 35 uA. Figure 7b shows the

process of current compensation and phase error change when the CP mismatch current is 10.2 µA before compensation.

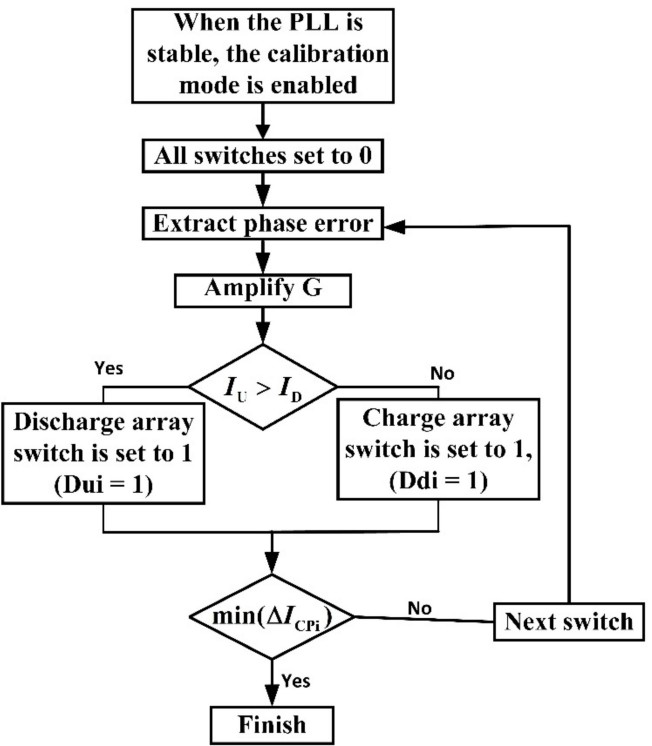

**Figure 6.** Flowchart for DAMC.

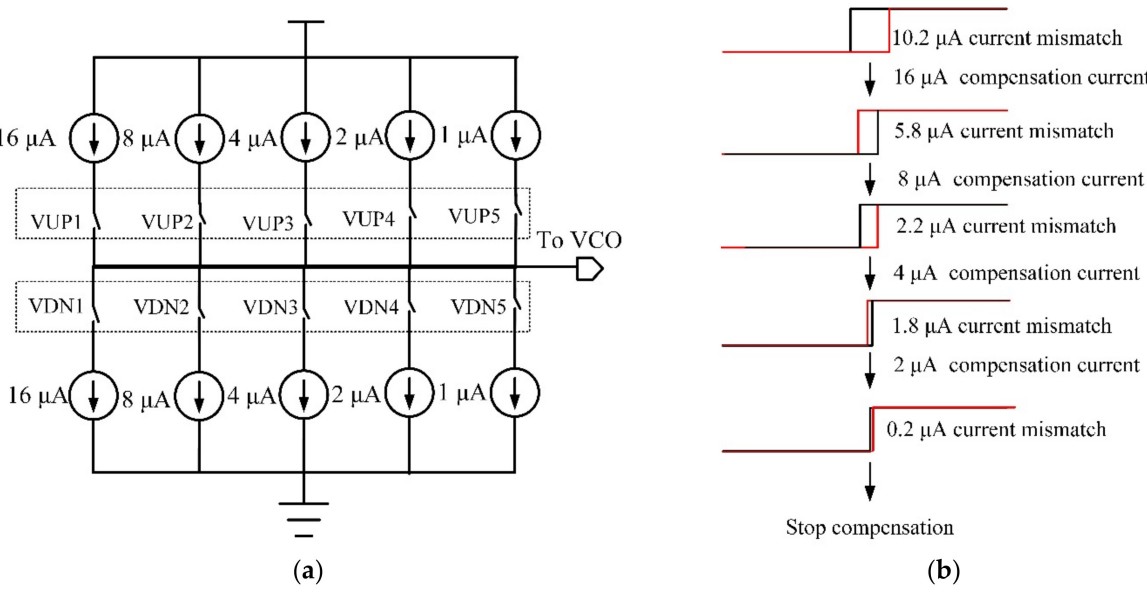

(**a**)

(**b**)

**Figure 7.** Examples of current mismatch compensation (**a**) compensation current source array; (**b**) the process of current compensation and the phase error change.

Figure 8 shows S-domain model of the PLL with DAMC. Regarding the charge pump current mismatch, the mismatch CP gain $K_{CP}$ is described as $\Delta K_{CP}$.

$$\Delta K_{CP} = \frac{\Delta I_{CP}}{2\pi} \tag{16}$$

After the DAMC operates, a new calibration gain $K_{CAL}$ is introduced. According to the process analysis diagram of the DAMC, calibration can be completed after at least M cycles. If DAMC completes the calibration in M cycles, the new calibration gain is given by

$$K_{CAL} = G \cdot \sum_{i=0}^{M} \varphi_{erri} \cdot K_{\Delta CPi} \qquad (17)$$

where $\varphi_{erri}$ represents the phase error extract at the i-th cycle and $K_{\Delta CPi}$ represents the compensation gain at the i-th cycle.

$$K_{\Delta CPi} = \frac{I_{CALi}}{2\pi} \qquad (18)$$

where $I_{CALi}$ represents the value of the calibration current in the i-th cycle. When $K_{CAL} = \Delta K_{CP}$, the gain error from the charge pump current mismatch is compensated.

At the same time, the compensation phase of DAMC ($\theta_{CAL}$)is written by

$$\theta_{CAL} = 2\pi \cdot \frac{G \cdot \Delta I_{CALi} \cdot T_d}{(I_{CP} - \Delta I_{CALi}) \cdot T_{REF}} \qquad (19)$$

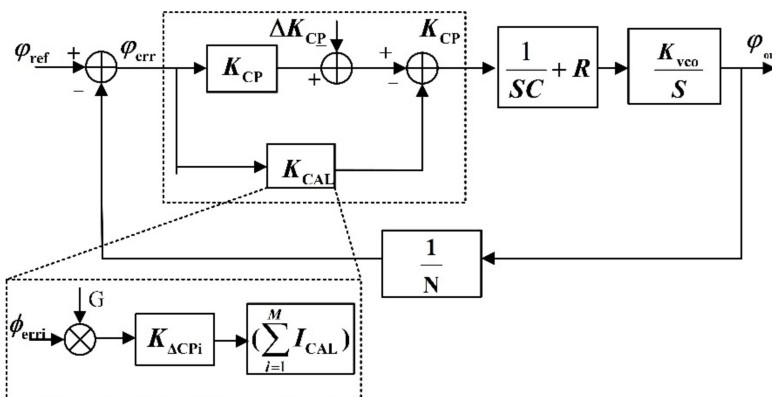

**Figure 8.** S-domain model of the PLL with DAMC.

## 4. Circuit Simulation and Test

The adaptive calibration TA proposed in this study had a wide dynamic range. Figure 9 shows the dynamic range simulation of the proposed TA. The TA gain G remains within 1.9 to 2.15 as the input changes from 10 ps to 110 ps.

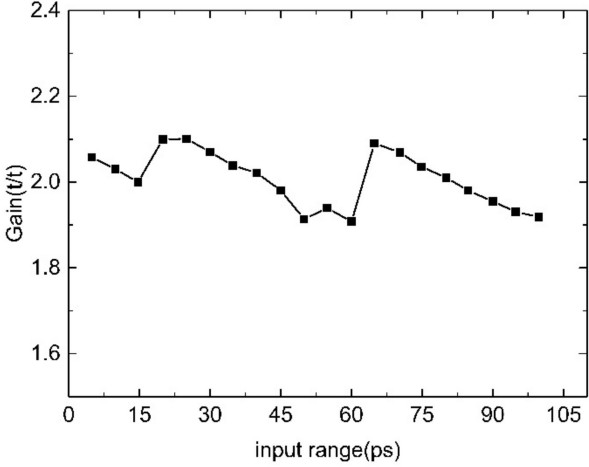

**Figure 9.** Dynamic range simulation of TA.

Five current sources were used for calibration based on the calibration method proposed in this study. Figure 10a shows one of the wave forms of the current mismatch calibration pulse (SAR). The simulation results show that 180ps static phase error is decreased to 30 ps, as shown in Figure 10b, and the VCO control voltage ripple and period jitter were reduced significantly. As shown in Figure 11, the P-P jitter of the 2 GHz clock is decreased from 29.2 ps to 2.6 ps.

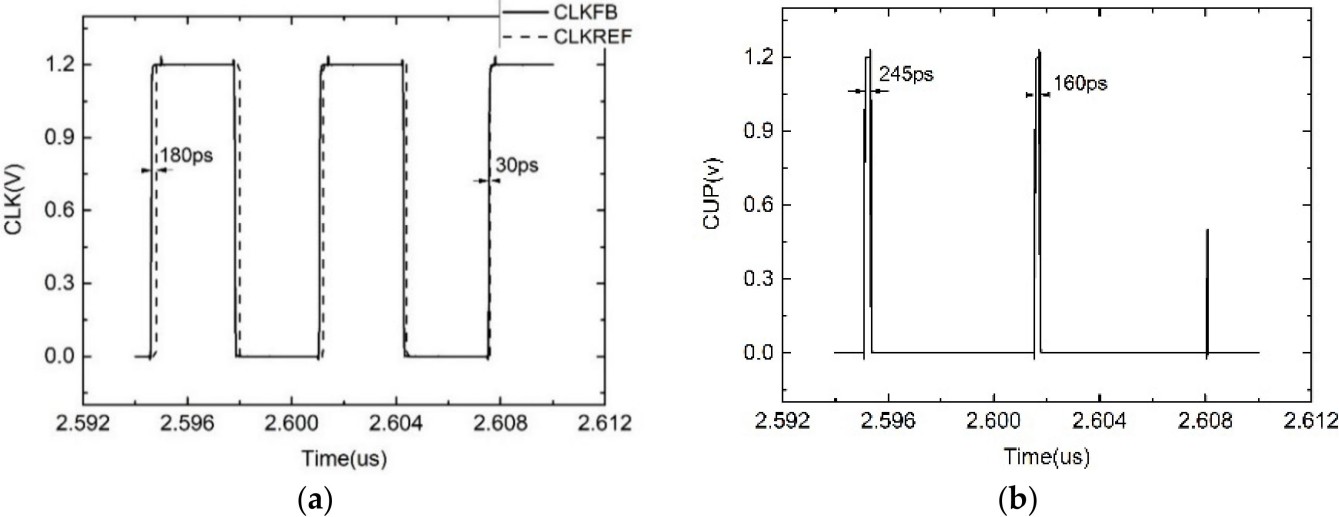

**Figure 10.** Simulation results (**a**) the PWC control pulse of SAR; (**b**) the phase error between CLKFB and CLKREF.

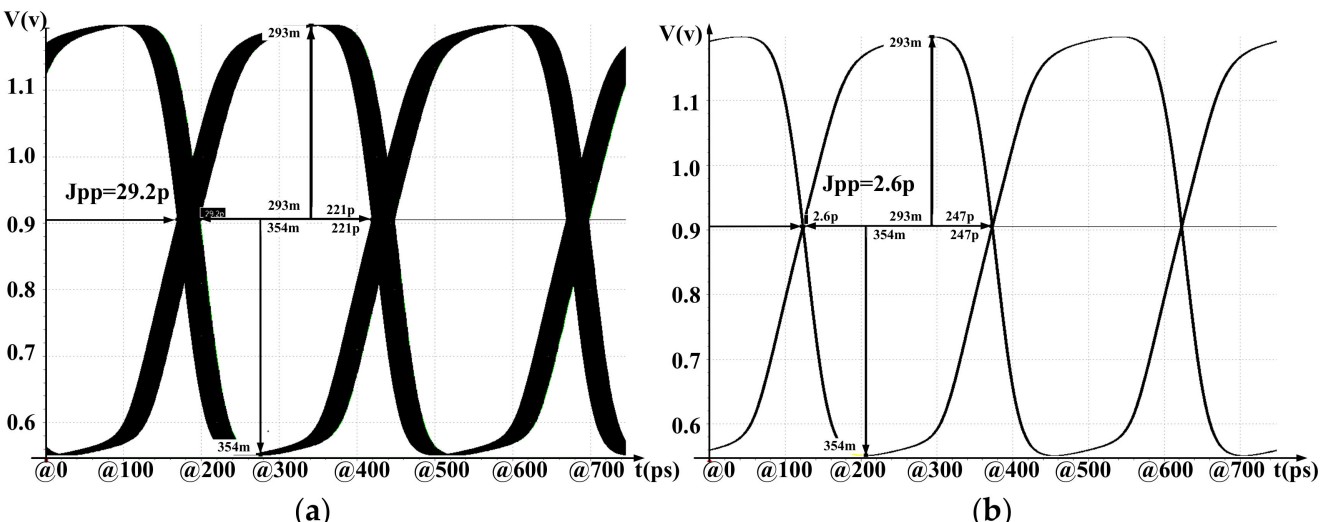

**Figure 11.** The eye diagram of the 2GHz VCO output (**a**) without calibration; (**b**) with calibration.

The PLL with DAMC was fabricated using 40 nm CMOS technology, as shown in the die photo of Figure 12. The PLL core occupies $420 \times 611 \ \mu m^2$. The DAMC occupies an area of $68 \times 52 \ \mu m^2$ (1.3% of the PLL area).

The static phase error also changed during adaptive calibration. The calibration test results when the reference clock is 50 MHz are shown in Figure 13, and the static phase error can be calibrated from 358 ps to 10 ps. Figure 14 shows that the RMS jitter of the PLL output is reduced from 4.91 ps at 2.5 GHz to 3.59 ps at 2.5 GHz. Table 1 shows the performance summary and comparison with prior PLL.

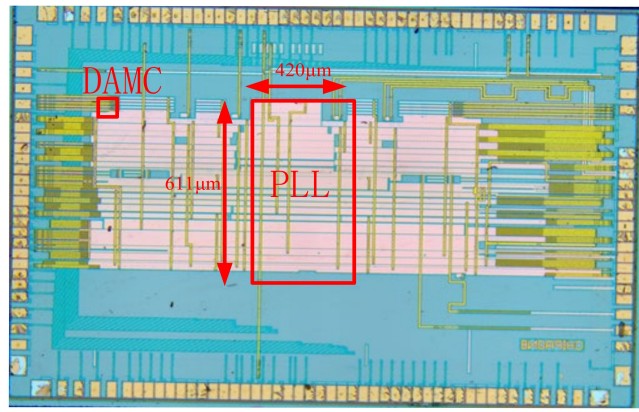

**Figure 12.** Die photo of the PLL with digital adaptive mismatch calibration.

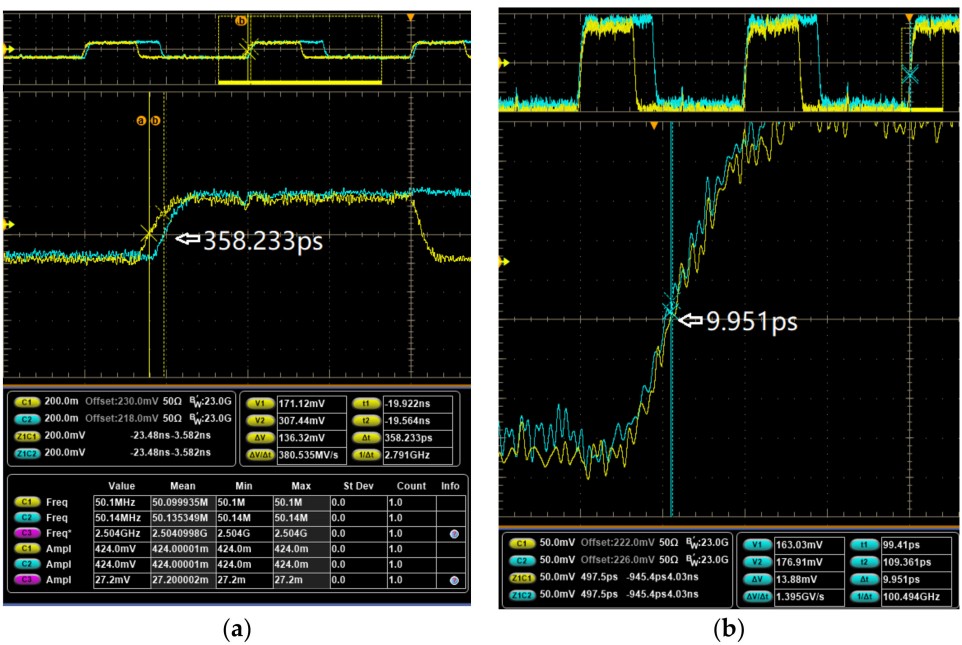

(**a**)　　　　　　　　　　　　　　　　(**b**)

**Figure 13.** Test results of the static phase error (**a**) without calibration; (**b**) with calibration.

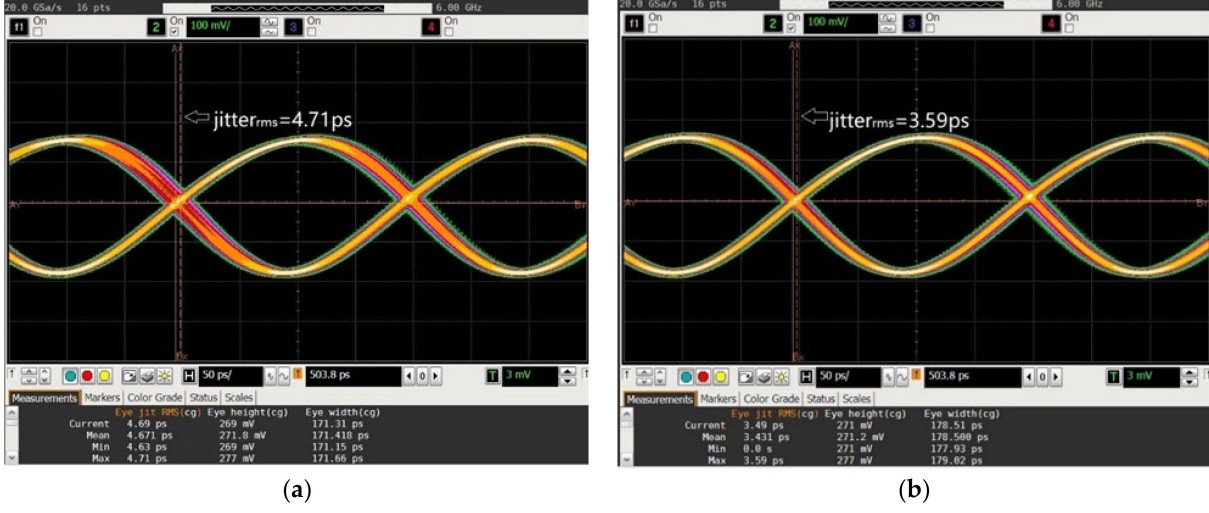

(**a**)　　　　　　　　　　　　　　　　(**b**)

**Figure 14.** Test results of the eye diagram at 2.5GHz (**a**) without calibration; (**b**) with calibration.

**Table 1.** Summary and comparison.

| | [4] | [6] | [7] | This Work |
|---|---|---|---|---|
| Output Frequency (GHz) | 0.8–3.2 | 3.2 | 0.5–5 | 1–4 |
| Topology | Ring | Ring | Ring | Ring |
| Reference Frequency (MHz) | 200 | 200 | 100 | 50 |
| RMS jitter(ps) | 2.52 at 2.4 GHz | 3.85 | 1.87 at 3.2 GHz | 3.59 at 2.5 GHz |
| $\frac{\Delta \text{RMS jitter}}{\text{RMS jitter}}$ (%) | N/A | N/A | 16.1% at 3.2 GHz | 23.7% at 2.5 GHz |
| $\frac{\Delta \text{static phase error}}{\text{static phase error}}$ (%) | N/A | N/A | 96% at 3.2 GHz | 97.2% at 2.5 GHz |
| Technology | 180 nm | 40 nm | 10 nm | 40 nm |

## 5. Conclusions

The paper presented a PLL with digital adaptive mismatch calibration to reduce the output clock jitter of the PLL. The proposed technique improves the calibration resolution with a TA and a new method, which minimized the algebraic sum of mismatch currents. In this study, we have analyzed an S-domain model of the PLL with DAMC and the perturbation effect of VCO output. The test results show that the proposed techniques reduced the reference phase error from 358 ps at 50 MHz to 10 ps at 50 MHz approximately, and the RMS jitter was reduced from 4.91 ps at 2.5 GHz to 3.59 ps at 2.5 GHz. The test proved that the technique is an effective way to compensate for the CP current mismatch and reduce the static phase noise in the background.

**Author Contributions:** Conceptualization, X.W. and L.H.; methodology, X.W.; software, L.H.; validation, X.W.; formal analysis, R.Y.; investigation, X.W. and R.Y.; data curation, X.W. and W.X.; writing—original draft preparation, X.W. and R.Y.; writing—review and editing, X.W., W.X.; and B.W.; visualization, X.W.; supervision, W.X.; and B.W.; project administration, X.W. and R.Y.; funding acquisition, X.W. and R.Y. All authors have read and agreed to the published version of the manuscript.

**Funding:** This work was supported by the National Nature Science Foundation (No. 62164003), Innovation Project of GUET Graduate Education (2022YCXS034), and the Foundation of Guangxi Key Laboratory of Wireless Wideband Communication and Signal Processing (No. GXKL06190110, No. GXKL06200131).

**Conflicts of Interest:** The authors declare no conflict of interest.

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
