# Peer review of "Design and Analysis of the Self-Biased PLL with Adaptive Calibration for Minimum of the Charge Pump Current Mismatch"

_electronics, doi:10.3390/electronics11142133_

Round 1

Reviewer 1 Report

The paper requires minor revision.

Author Response

Dear Reviewer,

Please find the revised version of manuscript (ID: electronics-1792409), entitled “Design and Analysis of the Self-biased PLL with Adaptive Calibration for Minimum of the Charge Pump Current Mismatch”.

Thank you for arranging a timely review for our manuscript. We have carefully evaluated the reviewers’ critical comments and thoughtful suggestions, responded to these suggestions point-by-point, and revised the manuscript accordingly. With regard to the reviewers' comments and suggestions, we’d like reply as follows:

Point 1: Cal-ibration changed to Cali-bration; con-vert changed to converter.

Response 1: Thinks,we have corrected word format.

Point 2: good linearity: repetition.

Response 2: Thinks, we have deleted repetition.

Point 3: Figure6: Please enlarge! It is unclear.

Response 3: Thinks, Figure 6 have been changed.

Point 4: Figure 10: please position the figure in the middle of the page.

Response 4: Thinks, Figure 10 has been changed to Figure 11. We have positioned Figure 11 in the middle of page.

Special thanks to you for your good comments.

Sincerely yours,

Xueming Wei, Renchuan Yin, Lingli Hou, Weilin Xu, Baolin Wei

Reviewer 2 Report

This article present a PLL design with digital adaptive mismatch calibration (DAMC) for improving charge pump current mismatch. Although some analysis and results are presented, some essential concerns need to be addressed listing below:

(1) Some descriptions are inaccurate, for example, in Page 2, it is mentioned the bandwidth and filter parameters are fixed in PLL design, this is not true, actually, even early PLLs have tunable capabilities on key parameters.

(2) In Fig. 8, the unit of y-axis is missing.

(3) The figures are vague. Also, there are many grammar problems requiring careful checking. The manuscript is not well written.

(4) The mechanism of the DAMC needs to be analyzed quantitatively with mathematical derivations on the calibration range, accuracy, etc.

(5) A comparison table with state-of-the-art PLL designs is needed.

(6) The motivation, novelty, and technical contributions of the proposed method need to be highlighted significantly.

Reviewer 3 Report

Here are my detailed comments, that in my opinion, can improve the manuscript:

1) line 21-22: "The impacts of mismatch currents on analog PLLs are the increase of static phase error, higher jitter and reference spur. " The target problem is enounced at the bennining of the introduction, however its description istoo brief, and although being an expert in ASIC design, the conclusions about the static phase error, jitter and spur are not obvious. Please elaborate more. Part of the first line of paragraph 2.2 can be moved in this part.

2) line 44: TA acronym not defined. 

3) Figure 2:  Separate in (a) and (b) the schematic and the plot. Define the quantities (voltages currents shown). Describe the schematic operational principle in the text, including the input/output quantities of the functional block. 

4) Figure 11: add reference scales 

5) General observations: (i) Figures should be referenced AFTER being mentioned in the text, not before; (ii) The "X@Y" is a shortform for "X at Y", not always accepted. The latter should be used.  

Round 2

Reviewer 2 Report

The concerns have been addressed in this revised version.